# An Algorithm to Minimize Energy Consumption and Elapsed Time for IoT Workloads in a Hybrid Architecture

**DOI:** 10.3390/s21092914

**Published:** 2021-04-21

**Authors:** Julio C. S. dos Anjos, João L. G. Gross, Kassiano J. Matteussi, Gabriel V. González, Valderi R. Q. Leithardt, Claudio F. R. Geyer

**Affiliations:** 1Institute of Informatics, UFRGS/PPGC, Federal University of Rio Grande do Sul, RS, Porto Alegre 91501-970, Brazil; jlggross@inf.ufrgs.br (J.L.G.G.); kjmatteussi@inf.ufrgs.br (K.J.M.); geyer@inf.ufrgs.br (C.F.R.G.); 2Faculty of Science, Expert Systems and Applications Laboratory, University of Salamanca, 37008 Salamanca, Spain; gvg@usal.es; 3COPELABS, Universidade Lusófona de Humanidades e Tecnologias, 1749-024 Lisboa, Portugal; valderi@ipportalegre.pt; 4VALORIZA, Research Center for Endogenous Resource Valorization, Polytechnic Institute of Portalegre, 7300-555 Portalegre, Portugal

**Keywords:** cost minimization model, energy consumption, Internet of things, mobile edge computing, scheduling algorithm

## Abstract

Advances in communication technologies have made the interaction of small devices, such as smartphones, wearables, and sensors, scattered on the Internet, bringing a whole new set of complex applications with ever greater task processing needs. These Internet of things (IoT) devices run on batteries with strict energy restrictions. They tend to offload task processing to remote servers, usually to cloud computing (CC) in datacenters geographically located away from the IoT device. In such a context, this work proposes a dynamic cost model to minimize energy consumption and task processing time for IoT scenarios in mobile edge computing environments. Our approach allows for a detailed cost model, with an algorithm called TEMS that considers energy, time consumed during processing, the cost of data transmission, and energy in idle devices. The task scheduling chooses among cloud or mobile edge computing (MEC) server or local IoT devices to achieve better execution time with lower cost. The simulated environment evaluation saved up to 51.6% energy consumption and improved task completion time up to 86.6%.

## 1. Introduction

An International Data Corporation (IDC) report predicts that there will be 41.6 billion IoT devices in 2025 with a potential for data generation up to 79.4 ZB [1]. IoT applications emerged with artificial intelligence, artificial vision, and object tracking in such a context that requires high computing power [2,3]. They usually rely on task processing offload and data storage to remote cloud computing (CC) data centers to boost processing time and reduce battery energy consumption [4]. Unfortunately, those remote servers are geographically located away from the end user and IoT devices, resulting in high latency due to delay and congestion over the communication channels [5,6,7]. Moreover, the use of centralized control (provider-centric) cannot deliver proper connectivity or even support computation closer to the edge of the network, thus becoming inefficient for highly distributed scenarios.

Mobile edge computing (MEC) can represent an option to increase the performance of CC applications, as it denotes a network architecture designed to provide low latency with adequate quality of service (QoS) to end users [8,9]. MEC relies on top high-speed mobile networks such as 5G to allow fast and stable connectivity for mobile devices and users. Thus, CC services can be deployed close to mobile devices, in the MEC layer, bringing processing and storage closer to cellular base stations [10].

Nevertheless, energy consumption remains a clear issue to be overcome on mobile device networks, such as MEC environments [11]. Most IoT sensors and mobile devices run on batteries with limited energy capacity [12]. Furthermore, IoT devices need to handle lots of data, which is also energy-consuming. Thus, reducing energy consumption in networks with IoT devices is a goal worth exploring.

The state of the art presents a set of studies that use MEC to offload tasks to offer local processing for IoT devices. Some works [4,13,14,15,16,17] have measured the energy consumption for data transmission or even using dynamic voltage and frequency scaling (DVFS) techniques. In contrast, this proposal enables a more detailed cost model, including energy and time consumed during processing and the cost of data transmissions. CC is also considered an option for processing when local resources are depleted, making the network more reliable in stress scenarios [18].

In this work, we explore the scheduling problems in edge computing environments, considering energetic consumption in a dynamic cost model to mitigate energy consumption in MEC environments. An algorithm called the Time and Energy Minimization Scheduler (TEMS) scheduling algorithm implements dynamic cost minimization policies correlating the system resource allocation with the more inexpensive cost for executing tasks. A simulator was developed for MEC evaluation with IoT devices and associated CC resources. The TEMS algorithm gathers data about the environment and associated energy and time costs to make decisions about the task scheduling. Code of the MEC Simulator is available at https://github.com/jlggross/MEC-simulator.

The main contributions of this work are:(i)The methodology covers a considerable number of energy and time metrics for task processing and data transmissions, including the accounting of idle CPU cores energy consumption;(ii)The CPU processing time and energy consumption optimization using DVFS technique;(iii)The scheduling policies consider task processing in the IoT device itself, in a local MEC server, and in a remote data center from CC at the same time.

The remainder of this paper is organized as follows. Section 2 discusses previous related work found in the literature. Section 3 lists the problems in MEC environments. Section 4 introduces the dynamic cost minimization model for the system with three different allocation policies, local processing in the IoT device, local processing in the MEC server, and remote CC processing. Section 5 introduces the TEMS heuristic scheduling algorithm designed to solve the cost minimization model of the system. Section 6 details the implementation and shows the results of the experiments using the TEMS scheduling algorithm. Finally, Section 7 presents the conclusions.

## 2. Related Work

IoT integrates several technologies for gathering data in the intercommunication world. Latency-sensitive applications need complicated processing such as that of time series analysis. However, IoT devices enable limited computing and energy resources to store large amounts of data and cannot perform complex task processing. The work proposed by [19] addresses the resource allocation and routing for IoT tasks that require efficient assignment in multicloud environments. The authors propose an energy-efficient, congestion-aware resource allocation and routing protocol (ECRR) for IoT networks based on hybrid optimization techniques.

Bi et al. [20] argue that task offloading leads to extra communication latency and energy cost. The work evaluated the offloading by finding an optimal offloading scheme that maximizes the system and seeks a balance between throughput and fairness.

Although MEC servers have been allowing intensive task computing in heterogeneous clouds, the data transmission over the Internet incurs high levels of access delay and jitter according to Zhao et al. [21]. This work minimizes MEC energy consumption and satisfies task processing delay requirements. The solution uses dynamic programming to minimize energy consumption by allocating bandwidth and computational resources to mobile devices.

The state of the art also shows that energy consumption decrease and response latency mitigation to applications in IoT environments are questions that are difficult to solve since the first edge computing architectures [22]. However, the strategic use of CC as a single alternative to task processing can add high latency due to the distance of IoT devices [23].

The proposals [11,24,25] use CC as an option for task execution. Other approaches use fog computing to allow local processing in IoT devices such as the works [14,26,27] or without applying the CC [28]. The MEC architecture is assessed in works of [4,11,13,25]. In contrast, TEMS is a three-layer architecture that combines MEC and CC added to local IoT computation. This approach also provides a cost model, with the energy and run time evaluations on the fly, including the data transmission costs.

As for the parameters used in third-party cost models, the energy consumption of task processing is used by all works mentioned. However, the energy consumption for data transmissions is shown in the works [4,13,14]. The processing time of tasks is evaluated in major of the works, except to [26,27] and the spent time on data transmissions is limited to studies of [4,11,13,14,25].

**Table 1 sensors-21-02914-t001:** Proposal comparison.

		Works														
		Wang et al. [4]	Sarangi et al. [11]	Zhang et al. [13]	Gedawy et al. [14]	Praveen et al. [19]	Bi et al. [20]	Zhao et al. [21]	Bui et al. [24]	Yu et al. [25]	Wan et al. [26]	Wu et al. [27]	Anjos et al. [28]	Naranjo et al. [29]	Mucchi et al. [30]	This work
**Architecture**	IoT device	X	X	X	X	X		X	X	X	X	X		X	X	X
	Fog (without MEC server)	X	X							X	X		X			
	MEC server			X			X	X						X		X
	Cloud		X			X	X	X	X	X			X	X		X
	Cluster with MicroCloud				X											
**Processing**	Local device	X	X	X	X	X		X	X	X	X	X	X		X	X
	Fog										X			X		
	MEC			X			X	X		X		X		X		X
	Cloud	X	X			X	X	X	X	X		X	X	X		X
	MicroCloud				X											
**Spent Time**	Task sending		X	X	X	X	X			X	X	X	X			X
	Code sending offloading		X	X	X		X	X		X	X	X				X
	Channel queue		X	X				X		X	X	X		X	X	X
	Task execution		X		X		X			X	X	X	X	X		X
	Data download			X	X		X	X				X				X
**Energy consumption**	Send of data and tasks	X	X	X	X		X		X	X			X	X		X
	Data download		X	X			X	X	X	X						X
	Data transport between devices	X		X		X			X					X		X
	Data aggregation													X		
	Dynamic frequency adjust		X	X				X	X	X						X
	Processor in running	X	X	X	X				X	X	X	X				X
	Sleep mode control														X	
	Idle processor	X		X	X				X				X			X
	Battery level			X	X				X	X		X		X		X
**Approach**	Consumption minimization	X	X	X		X	X	X	X	X	X	X		X		X
	Execution in the master node				X					X		X	X			X
	Optimization problem	X	X	X	X	X		X		X	X	X	X			X
	Latency minimizing			X	X		X	X		X	X		X	X		X
	Channel capacity control														X	
	Simulation		X	X			X	X		X	X	X	X	X		X

5G networks introduce designs and adaptable support to new applications. However, it requires latency management, high energy efficiency, and long-range communication support for IoT-based applications [31]. Mucchi et al. [30] propose to add a physical layer to the burst data transmission management with the insertion of a zero-energy symbol for the wireless transmission when there is the send of discontinuous data (silence periods). As a result, the system saves energy consumption from IoT devices. In this approach, the silence produces a fine-grain granularity to energy management. The algorithms optimize power consumption between processing and energy spent on transmission.

On the other hand, the energy consumption of the equipment in the idle state is measured exclusively in [26,27]. Models that include the battery level of the IoT devices are only [4,13,14]. Our proposed model uses the DVFS technique similar to [16,17] to allow both the dynamic minimization of energy and execution time during task processing.

Naranjo et al. [29] propose an energy-maximizing solution to prolong the aliveness of the wireless sensor networks. The proposal uses a prolong stable election protocol (P-SEP) in a fog infrastructure to decrease energy consumption. An algorithm considers the distance between a cluster head (grouping of IoT sensors) and fog nodes to achieve load balancing.

Gautham et al. [32] analyzed an architecture to evaluate the code/decode of the communication channel. In particular, TEMS determines the better communication channel between IoT devices and MEC servers. Therefore, the contribution of channel coding/decoding and error correction are included in the choice of the channel with the lower cost.

All these variables, the execution time, energy consumption, architectures, and approaches are consolidated in Table 1. In our proposal, a monitor for battery levels of IoT devices allows rational energy consumption. Two task policy types, critical (to deal with deadline constraints) and regular (to deal with common executions) are explored in the scheduling algorithm.

## 3. Problem Statement

A single cost model needs to evaluate deployment with all used variables for data transmissions, such as time and energy consumption. It must determine the local to task execution among MEC, fog, or cloud environments. This is a multiple-objective optimization problem. Therefore, finding a minimal cost to all these system variables is an NP-hard problem.

The solution must consider the energy consumption, such as the energy to send tasks, energy consumption to data download, energy for the task processing, energy cost with CPU idle, and battery level. It also requires estimating the execution time to variables, such as time spent on task transmission, wait time in queues, task runtime, and time spent on downloads. Simultaneously, the system must choose one among three distinct environments to produce the best performance considering energy optimization.

Thus, the computation to solve these issues is hard to model. Deciding between three different environments is another complex task due to needing to produce task distributions with adequate performance and energy saving. We propose a dynamic solution in real-time for each task using an integer linear programming (ILP) optimization to achieve this challenge.

## 4. Model to Minimize Cost Dynamically

This section introduces the model to mimize cost dynamically.

### 4.1. Architecture and Task Processing Flow

Figure 1 exhibits the architectural scheme on three decoupled layers under a bottom-up view:IoT Layer (L1): IoT device layer generates application tasks. These devices have a limited processing capability and operate with batteries;MEC Layer (L2): MEC server layer has a restricted number of CPUs and less processing capability than the CC environment. MEC servers are closer to the IoT devices, producing smaller communication delays;CC Layer (L3): CC data centers compose this layer. These servers have high processing capability, are geographically distributed, and are located far from the IoT devices. They also add high network latency due to data transmission with more communication hops, if compared to other layers.

**Figure 1 sensors-21-02914-f001:**
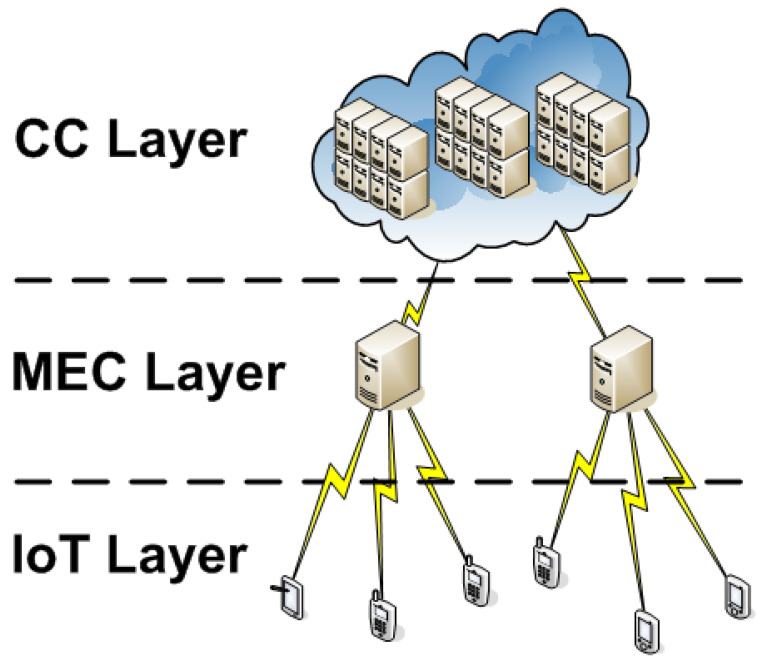
System architecture.

The model associates the cost in terms of energy consumed and elapsed time for the allocation policy of each layer, taking into account task processing and data transmissions costs. The DVFS technique is used to calculate processing costs, proving the best pair of CPU core voltage and CPU core operating frequency that reduces total cost.

Finally, the TEMS scheduler seeks the best cost among all three allocation policies and selects the lowest one. The scheduler decides between MEC and CC layers to offload a task. Otherwise, the processing takes place on the device itself.

The rest of this section covers an extension of the cost model of our previous work shown in [33]. First, the assumptions about the network and the architecture components are introduced. After that, the cost models for local computing in the IoT device, local computing in the MEC server, and remote computing in the cloud are shown. Finally, the individual costs are combined into a final equation that represents the total cost.

Table 2  summarizes the notation used throughout the model definitions of network, local IoT computing, MEC server, and cloud computing.

### 4.2. Network Model

The network is composed of mobile IoT devices, MEC servers, and a cloud provider. The wireless links determine the communication channels between IoT devices and MEC servers, as in Figure 1. The system network has a finite set D={1,2,3,…,d} of mobile IoT devices, S={1,2,3,…,s} of local MEC servers, and of wireless communication channels $W_{C}=\left\{1,2,3,\ldots ,w\right\}$. Each scheduled task has a set of the available wireless channels ($H=\{h_{1},h_{2},\ldots ,h_{a}\}$) from the IoT device to MEC and from the MEC to CC with its correspondent bandwidth. The TEMS algorithm will choose a channel with a lower energy consumption cost. If the task runs in the local device, it does not have an associated channel. Coding and decoding costs are built into the channel like in the approach [32].

### 4.3. General Energy Consumption

Equation (Equation 1) computes the energy consumption based on the dynamic power consumed during the execution. The potency is a relation $\propto CV^{2}f$, defined in Liu et al. [34]. Each device type $D_{T}$ has a frequency and a commutative capacitance *C* associated with a core *k* in the processor, which depends on the chip architecture [25]. The potency is defined in Equation (Equation 2).

A system has a total of A={1,2,3,…,a} tasks. Each task *i* is associated to a tuple $A_{i}=\left(C_{c_{i}},sc_{i},d_{i},t_{i}\right)$ composed of CPU cycles ($C_{c}$) needed to conclude an execution. The tuple includes the source code (sc) offloading from IoT to MEC, the input data (*d*), and the deadline (*t*) associated with the task. Offloading is an advantage in industrial applications to reduce the congestion of data transmission and save energy consumption [13].

The CPU cycles is a task property. The total execution time of a task is calculated based on total cycles $C_{{cT}_{i}}$ [35] where i∈A to a CPU core j∈PL in Equation (Equation 3). Each local IoT device or MEC server can process zero or more tasks. The deadline associated represents if a task is normal (t=0) or if it is critical (t>0).
(1)$$E_{i,D_{T}\left(k\right)}=P_{i,D_{T}\left(k\right)}*T_{i,D_{T}\left(k\right)}$$
where,
(2)$$P_{i,D_{T}}=C_{D_{T}\left(k\right)}*V_{D_{T}\left(k\right)}^{2}*f_{D_{T}\left(k\right)}$$
(3)$$T_{i,D_{T}}=\frac{C_{{cT}_{i}}}{f_{D_{T}\left(k\right)}}$$

### 4.4. Local Computing in the IoT Device

Each mobile device has a respective number of CPU cores ($PL_{j}=\left\{pl_{j,1},pl_{j,2},\ldots ,pl_{j,n}\right\}$). The energy consumption is computed in Equation (Equation 4) based on a total number of CPU cycles ($Cc_{i}$), operating frequency ($f_{local,j,k}$), voltage supply ($V_{local,j,k}$), and on the effective commutative capacitance ($C_{local,j,k}$) of each core. Equation (Equation 4) computes the local dynamic energy consumed for the IoT device in each task.
(4)$$E_{i,local}=P_{i,local}*T_{i,local}$$
where,
(5)$$T_{i,local}=\frac{C_{{cT}_{i}}}{f_{local,j,k}}$$
(6)$$P_{i,local}=C_{local,j,k}*V_{local,j,k}^{2}*f_{local,j,k}$$

Considering battery level and latency as model constraints, a device $D_{j}$ must decide whether it is more appropriate to process the task locally or remotely. As the battery level is a critical factor in the decision, the system will appreciate a policy that reduces energy consumption. The local cost of one task *i* is expressed in Equation (Equation 7).
(7)$$\begin{array}{c} Cost_{i,local}=u_{localT}*T_{i,local,total}+u_{localE}*E_{i,local} \end{array}$$

The coefficients $u_{localT}\in \left[0,1\right]$ and $u_{localE}\in \left[0,1\right]$ are weightings, where $u_{localT}+u_{localE}=1$. These variables represent a trade-off between execution time and energy consumption and minimize one of the costs, according to Wang et al. [4]. The DVFS associated overhead rate is between 0.02% to 2% in the best- and worst-case scenarios, according to [16]. The cost overhead for this approach represents 2% in our model, built into these trade-off coefficients.

### 4.5. Local Computing in the MEC Server

A local MEC server can have several CPU cores. Thus, the CPU cores available on a local server $S_{j}$ are given by $PS_{j}=\left\{ps_{j,1},ps_{j,2},ps_{j,3},\ldots ,ps_{j,n}\right\}$. Each core $ps_{j,k}$ has an operating frequency ($f_{mec,j,k}$), an effective commutative capacitance ($C_{mec,j,k}$), and a supply voltage ($V_{mec,j,k}$).

IoT devices and MEC servers cause mutual interference between each other ($I_{i}$) because they use the same wireless channel. Thus, the data transfer rate ($r(h_{i})$) to offload task *i* to the channel ($h_{i}$) attenuates according to Shannon’s formula [25]. The data transfer rate is determined in Equation (Equation 8) and the mutual interference between wireless channels bandwidth (*B*) is computed in Equation (Equation 9).
(8)$$r\left(h_{i}\right)=B*\log_{2}\left(1+\frac{p_{j}*g_{\left(S_{l},j\right)}}{N+I_{i}}\right)$$
(9)$$I_{i}=\sum_{n\in A|\left\{i\right\}:h_{n}=h_{i}}p_{j^{\prime }}*g_{\left(S_{l}^{\prime },j^{\prime }\right)}$$

For Equation (Equation 8), the variable $p_{j}$ is the transmission power of a mobile device *j* during offloading task *i* to the local server, and *N* is the power of the thermal noise of the wireless channel. $g_{\left(S_{l},j\right)}$ is the channel gain between the local MEC server and the mobile device.

In the local server, data and source code need to be sent to the application processing, and the generated results must be sent back to the origin. Thus, the time required for an IoT device to sent data (Equation (Equation 10)) and after to download the results (Equation (Equation 11)) from the local server can be computed as:(10)$$T_{i,mec-up}\left(h_{i}\right)=\frac{sc_{i}+d_{i}}{r_{i}\left(h_{i}\right)}$$
(11)$$T_{i,mec-down}\left(h_{i}\right)=\frac{d_{i}^{\prime }}{r_{i}\left(h_{i}\right)}$$

The total time required to complete the task execution in the local server considers the send (Equation (Equation 10)), the download (Equation (Equation 11)), and the task execution time in the MEC server calculated in Equation (Equation 5). The total time for a MEC server is given as in Equation (Equation 12).
(12)$$\begin{array}{cc} T_{i,mec,total}= & T_{i,mec-up}\left(h_{i}\right)+T_{i,mec}+\\ \; & T_{i,mec-down}\left(h_{i}\right) \end{array}$$

The energy spent for the data communications from the local MEC server is also calculated by (Equation (Equation 1)), which can be either the time to sent (*mec-up*) or download (*mec-down*) data. Furthermore, the dynamic energy consumed by the MEC server is calculated in the same fashion as that in the IoT device. Equation (Equation 13) gives the total dynamic energy consumption.
(13)$$\begin{array}{cc} E_{i,mec,total}= & E_{i,mec-up}\left(h_{i}\right)+E_{i,mec}+\\ \; & E_{i,mec-down}\left(h_{i}\right) \end{array}$$

Moreover, the cost computation for the local server is expressed in Equation (Equation 14).
(14)$$\begin{array}{cc} Cost_{i,mec}= & u_{mecT}*T_{i,mec,total}+\\ \; & u_{mecE}*E_{i,mec,total} \end{array}$$

### 4.6. Remote Computing in the Cloud

The CPU cores in CC are not distinguished because they are a single shared resource comparable to a CPU processor. It is not really a device. The CC equations are analogous to those of the local MEC server. Data transference between MEC and CC is composed of both the elapsed time and consumed energy to produce a total cost. The elapsed time is expressed by (Equations (Equation 15) and (Equation 16)), while consumed energy is expressed in Equations (Equation 17) and (Equation 18).
(15)$$T_{i,cloud-up}=\frac{s_{i}+d_{i}}{r}$$
(16)$$T_{i,cloud-down}=\frac{d_{i}^{\prime }}{r}$$
(17)$$E_{i,cloud-up}=p_{wireless}*T_{i,cloud-up}$$
(18)$$E_{i,cloud-down}=p_{wireless}*T_{i,cloud-down}$$

Note that in Equations (Equation 15) and (Equation 16), *r* is not dependent on $h_{i}$, because transmissions between MEC and CC are done on fiber optic cables, and there is no mutual interference effect attenuating the data transmission rate. CC processing time (Equation (Equation 19)) and dynamic energy consumed (Equation (Equation 20)) are calculated the same way for MEC servers. The total elapsed time and total energy consumption for CC are as follows.
(19)$$\begin{array}{cc} T_{i,cloud,total}= & T_{i,mec-up}\left(h_{i}\right)+T_{i,cloud-up}+T_{i,cloud}+\\ \; & T_{i,cloud-down}+T_{i,mec-down}\left(h_{i}\right) \end{array}$$
(20)$$\begin{array}{cc} E_{i,cloud,total}= & E_{i,mec-up}\left(h_{i}\right)+E_{i,cloud-up}+E_{i,cloud}+\\ \; & E_{i,cloud-down}+E_{i,mec-down}\left(h_{i}\right) \end{array}$$

Finally, the cost to run a single task *i* in the cloud is given in Equation (Equation 21).
(21)$$\begin{array}{cc} Cost_{i,cloud}= & u_{cloudT}*T_{i,cloud,total}+\\ \; & u_{cloudE}*E_{i,cloud,total} \end{array}$$

The idle energy cost of CC is not considered, since the CPU offer is virtually infinite. Therefore, it does not make sense to account for this cost, which would cause the system to have equally infinite cost.

For every task *i*, the minimum cost is chosen between all three allocation policies, one from each layer, as in Equation (Equation 22).
(22)$$\begin{array}{c} Cost_{i}=min\left(Cost_{i,local},Cost_{i,mec},Cost_{i,cloud}\right) \end{array}$$

The total system cost, represented by Equation (Equation 23), is equal to the sum of all task costs plus the sum of idle energy for IoT devices and MEC servers.
(23)$$Cost_{system}=\sum_{i=1}^{A}Cost_{i}+\sum_{i=1}^{A}E_{local_{i},idle}+\sum_{i=1}^{A}E_{mec_{i},idle}$$

### 4.7. Model Constraints for IoT Device Battery

A healthy battery level is essential to the proper operation of IoT devices. If the battery level $B_{j}$ of an IoT device *j* is below a lower safety limit (LSL), task allocation on the device is disabled to keep the device alive with the remaining battery. If $B_{j}$ reaches zero, all tasks generated by device *j* are canceled. Therefore, to prevent this from happening, the cost equations are subject to the following constraints: $B_{j}>E_{i,local}$, $B_{j}>E_{i,mec-up}\left(h\right)$. These constraints are considered in the scheduling algorithm.

## 5. The TEMS Algorithm

The heuristic of the scheduling algorithm for time and energy minimizing was developed to execute a dynamic cost minimization model with reduced computational cost. Algorithm 1 exhibits the steps of TEMS. Step 1 is the configuration detection round that forms a data set of IoT devices, MEC servers, and configuration of communication channels. The battery levels of the IoT devices are collected, and the LSL is established. The algorithm regards the number of cores into CPU available in each IoT device and MEC server, the operating frequency, and operating voltages. This process could also occur in CC data centers, but the number of CPUs is expected to be unlimited.

A loop from step 2 to step 4 repeats until all tasks are distributed across the processing infrastructure. The scheduler observes hardware conditions, energy consumption, and performance of application execution.
**Algorithm 1:** TEMS.
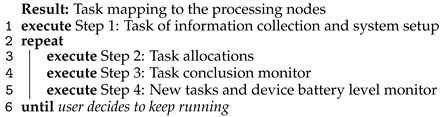


Step 3 monitors the task completion status. When one task was completed, the CPU core resources are released to turn available for new allocations in step 2. However, tasks that use CC resources do not need to release them since CC is supposed to have unlimited resources, absorbing any number of tasks. Task cancellation may occur if the elapsed time is higher than the deadline or if the IoT device runs out of battery.

Finally, in step 4, the battery level from each IoT device is collected, and after that, it creates new tasks again. Execution continues as long as tasks are being created.

Algorithm 2 details step 2 from task allocations, which is the task allocation decision-making process of the scheduler. Here, tasks are first classified between critical and regular.
**Algorithm 2:** Task allocations
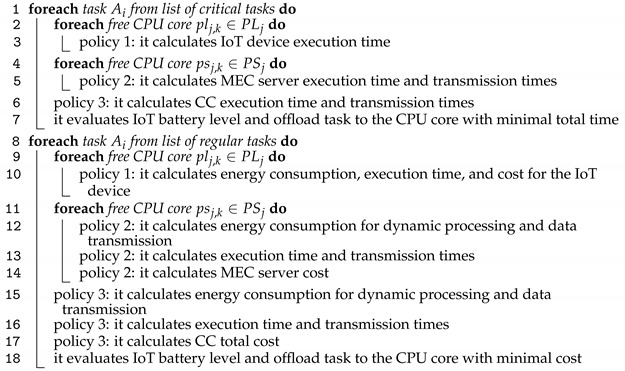


Three policies are defined for task execution: Policy 1, the tasks are executed in local IoT devices; Policy 2, the tasks are offloaded from IoT devices to the MEC server and executed in the MEC server; Policy 3, the tasks are executed in the cloud.

The time and energy consumption for task processing on the different CPU cores of the network is calculated, as well as the time and energy consumption of the data transmissions for MEC servers and CC data centers. Critical tasks are the first to be scheduled due to the execution deadline. The tasks are launched from the lower to higher deadline and allocated to produce the lowest total elapsed time considering latency and the channel bandwidth availability. The regular tasks are ordered by creation time and allocated by the minimum total cost. The battery level of IoT devices is continuously evaluated in lines 7 and 18 to check if the energy constraints are respected.

### Algorithm Complexity Evaluation

The algorithm complexity analysis considers the four steps in Algorithm 1. The task of information collection and system setup occur a single time in the system setup. This step identifies “*n*” mobile devices added to the network, and it has “*m*” processor cores. There are a total of “*n*” local MEC servers with the “*m*” core processors and a number of “*n*” wireless network channels to “*n*” tasks with a tuple of four variables each. The algorithm must choose among “*n*” possible options with three variables each and nine coefficients to the cost equation. Thus, for step 1, the complexity is defined in Equation (Equation 24).
(24)$$\begin{array}{c} nm+nm+n+4n+3n+9=\\ 2nm+8n+9=O(nm) \end{array}$$

However, the smartphone currently has a limit of eight cores. Additionally, simple IoT devices, for instance, Arduino Mega 2560, have a single core. MEC servers can be composed of up to five Raspberry Pi IV with four cores. Thus, the processor cores number is less than the amount of then IoT devices, i.e., m<<n, and if *m* is a mensurable and a finite number, then it is reasonable to think that m≈k and in this scenario O(nm)=O(kn)≈O(n). Therefore, in step 1, the complexity is O(n).

In step 2, the task allocation has a sort function with O(nlog(n) complexity in the worst case and O(n) in the best case. A seek is executed two times among *n* tasks into *n* local devices and MEC servers to achieve the lower execution time and energy consumption. This task has $O(n^{2})$ complexity. The cloud allocation tasks have O(1) complexity. The TEMS algorithm was developed with Python programming, and the Python sorting executes three times. Thus, the complexity for step 2 is described in Equation (Equation 25).
(25)$$\begin{array}{c} 3\;O\left(nlog\left(n\right)\right)+2\;\left[O\left(n^{2}\right)+O\left(1\right)\right]=\\ 2\;O\left(n^{2}\right)+3\;O\left(nlog\left(n\right)\right)+2\;O\left(1\right)=O\left(n^{2}\right) \end{array}$$

Thus, the step 2 has a complexity $O(n^{2})$.

Step 3 has *n* interactions of simple tasks, so O(n), and step 4 seeks the battery level in *n* IoT devices, i.e., O(n).

Hence, considering all TEMS algorithm steps and exchanging these steps by respective individual complexity as in Equation (Equation 26),
(26)$$O\left(TEMS\right)=O\left(n\right)+O\left(n^{2}\right)+O\left(n\right)+O\left(n\right)=O\left(n^{2}\right)$$

Therefore, the algorithm complexity is $O(n^{2})$.

## 6. Evaluation

This section shows the evaluations and explains the simulation details and the different experimental scenarios used.

### 6.1. Simulated Hardware and Software Stack

The simulated environment was designed with low, mid-range, and high processing power devices for IoT, MEC, and CC layers, respectively. For IoT devices, we chose Arduino Mega 2560, with five operating frequencies and corresponding supply voltages for DVFS. The MEC servers were simulated on top of 5 Raspberry Pi 4 Model B boards, each board with a Quad-core Cortex-A72 1.5GHZ (ARM v8) 64-bit, summing a total of 20 CPU cores per server. These CPU cores have three operating frequencies and corresponding supply voltages. Table 3 specifies the voltage–frequency pairs and the capacitance of the underlying hardware architecture of IoT and MEC devices. These combined values are used to calculate the power consumed by a device according to the selected values.

For CC, we chose data centers with Intel Xeon Cascade Lake processors of 2.8 GHz per CPU core, reaching up to 3.9 GHz with *Turbo Boost* on (Technical information can be found in the data sheets of the electronic components). Here, there are no voltage or capacitance variables. Instead, the resulting power is used, 13.85 Watts and 24.28 Watts, for configuration with and without Turbo Boost, respectively.

The network throughput was configured to achieve up to 1 Gbps speed and latencies to 5 ms, for both 5G and fiber optic communications [36,37]. The simulated applications are two vehicular applications that were described originally in Jansson ’s Ph.D. thesis [38]. Application 1 represents the image recognition of vehicle registration plates on roads. It has a high workload and a high task creation rate. Application 2 is a vehicle-to-vehicle communication to avoid car collisions. It is a critical-mission application with a low workload, low data, and a hard deadline. Because of this, the task generates rates are faster, in comparison with Application 1. Application 2 creates more tasks than Application 1 for the same interval time, but each task has lower processing requirements. Table 4 shows the characteristics of each application.

### 6.2. Experiments and Results

The tested scenarios evaluated the size of data entry and results, task generation rate, deadline of critical tasks, level of batteries for IoT devices, and use of DVFS. The main goal is to see as the TEMS algorithm responds to energy consumption and execution overall behavior. The results are discussed below.

#### 6.2.1. Use of MEC Servers

This experiment evaluates behavior when varying the number of MEC servers in the system. Application 1 is used, and the workload was configured according to the description in Table 4. The tested scenario has 500 tasks distributed to 100 IoT devices in two different cases, one with a single MEC server, in Figure 2a,c and another with two MEC servers, in Figure 2b,d. Figure 2 depicts the results for the execution of Application 1 and Application 2 in both cases with $10\times 10^{6}$ CPU cycles. The *x*-axis shows the execution time in seconds, and the *y*-axis indicates the number of tasks.

The energy and time coefficients were set, respectively, to 4/5 and 1/5, that is, a high weight was given to the energy consumed so that it could be minimized. In Figure 2 from plot Figure 2a to plot Figure 2b and from plot Figure 2c to plot Figure 2d, there is an increase in the number of MEC servers, from one to two, which made fewer tasks be to allocated in the CC layer. This positively impacts the total energy consumed because tasks running in the MEC layer demand less energy when the workload is higher. However, when the workload is composed of small tasks with a high transfer rate, the scheduler tends to maintain all tasks nearest from devices due to deadline restrictions.

Table 5 shows the relationship between the number of MEC servers related to energy consumption. When comparing cases A and B with a third case C with no MEC servers, the reduction in energy consumption for case A was 42.51%, while for case B 44.71%. In case C, tasks are just offloaded to the Cloud, adding too much energy consumption to the system. Thus, the use of MEC servers helps the system to lower the total energy consumed.

With Application 2, which has lower workload compared to Application 1, the allocation profile changed. Most allocations took place on the device itself, regardless of the number of MEC servers. The cause to this phenomenon is due to the low processing workload of Application 2. The hardware of IoT devices presents higher energy consumption per CPU cycle. However, it does not require data transmissions, which add energy cost and elapsed time to the system. Thus, for a small processing workload, IoT devices are the first allocation option.

#### 6.2.2. IoT Device Battery Energy Consumption

Figure 3 shows an experiment executed for Application 2 with 10,000 tasks, 100 IoT devices, and 2 MEC servers. Initially, Figure 3a shows the tasks are allocated according to their type. The *x*-axis shows the time in seconds, and the *y*-axis indicates the number of tasks. Regular tasks run on the IoT device itself due to the lower cost among all allocation policies, while critical tasks run on the server, as the total time is reduced compared to the IoT device, even though the energy cost is higher. Thus, tasks are distributed for local processing in the IoT device and in the MEC server.

Figure 3b represents the battery energy consumption of one IoT device of the system. The *x*-axis indicates the time in seconds, and the *y*-axis shows the battery energy consumption in Joules. At around 15 s, the battery energy consumption level reaches the LSL. It corresponds to 10% of the maximum battery capacity. From this moment forward, TEMS no longer allows tasks to run on the IoT devices, causing a sudden increase in the number of allocations to the MEC server for the newly created tasks.

Our analysis indicates that low battery levels quickly reach LSL and make IoT devices unavailable for processing. High computational workloads also negatively affect the battery level. Therefore, a battery with a healthy energy level and adequate task processing workloads allows the allocation to be performed on the IoT device, without making it unavailable due to lack of battery, contributing to total cost reduction.

### 6.3. Accuracy Evaluation of Energy Model

Analytic analysis of energy model accuracy is exhibited in Figure 4 considering the experiment of Figure 3a. The *x*-axis shows the execution time in seconds and the *y*-axis shows the accuracy. This analysis evaluates a particular case where the battery of IoT is ideal, that is, the battery of the IoT devices is infinite. The execution profile of the Figure 3b is compared with the ideal system, taking into account that the scheduling mechanism must choose the lower cost for execution time.

When the computational resources of IoT devices are busy, the system only can choose freed resources, even if they are more expensive. Thus, the scheduler can not more maintain a lower cost for the system.

#### 6.3.1. Variation of Input Data Size

This experiment evaluates how the costs of each allocation policy change according to the data size for tasks from Application 1, shown in Figure 5. We built a simulation with 500 tasks, 100 IoT devices, two MEC servers, and energy cost coefficient configured to 4/5. The *x*-axis shows the input sizes. Figure 5a shows 3.6 MB, 36 MB and 362 MB, and Figure 5b shows 3.6 GB. The *y*-axis indicates the cost of policies.

As shown in Figure 5, when the data entry size increases, MEC and CC policies have cost increments. The IoT execution cost in devices remains the same, as no data transmissions are carried out. When data entry size scales, allocation policies that require data transmissions become costly, and allocation on the IoT device itself turns increasingly advantageous. This increase in cost for MEC and CC policies is quite evident in Figure 5b, for inputs of 3.6 GB, even the cost scale had to be adjusted to represent the values better. Therefore, it is crucial to design applications so that data transfers over the network are not too large per task, avoiding high data transmission costs. An approach to do this is to create more tasks with lower data size.

We also designed two other cases, one with 5000 tasks and 362MB per task and another with 500 tasks, Figure 5a, and 3.6 GB per task, with roughly 1.8 TB in total each. The system energy consumption and the total elapsed time for the 500 tasks case were 59,160.92 Joules (J) and 14,515.21 s. For the experiment with 5000 tasks, the costs were 45,011.49 J and 10,305.94 s, that is, a decrease of 23.92% and 29%, respectively. Therefore, tasks should preferably not be super data-intensive, if dependent on MEC or CC, as data transmissions add additional energy and time expenses.

#### 6.3.2. Impact of Energy and Time Coefficients in the Schedule Policy Choices

This experiment used Application 2 in four different scenarios. Each case with 500 tasks, 100 IoT devices, and one MEC server. The energy coefficients were set to 1/5, 2/5, 3/5, 4/5 and the time coefficients to 4/5, 3/5, 2/5 and 1/5.

Table 6 lists the minimum costs identified by the system task scheduler for each case. The lowest calculated cost was the same for C2 and C3, with MEC as an offloading option. C1 case had the lowest calculated cost among all coefficient pairs. In these three cases the time coefficient had high values, and MEC was chosen because task execution got the lowest processing times. For C4, the allocation took place on the IoT device itself, with DVFS configured at 8 MHz and 4 V. Now, energy has a high-value coefficient, which made the scheduler choose the policy that provided the lowest energy cost, reducing total cost.

To reduce energy consumption, the best option is to use 4/5 as an energy coefficient. With this configuration, the minimization of energy consumption is prioritized, saving up to 51.6% compared to the other cases. Alternatively, to reduce task completion time, coefficients from C1, C2, and C3 cases are better, with a reduction of up to 86.6% compared with the C4 case. For C1 to C3 cases, a considerable reduction in total elapsed-time was perceived. It because the task launches in the IoT devices caused the increase the time execution.

#### 6.3.3. Impact of Task Generation Rate

This experiment executes four scenarios, with task generation rates of 0.05, 0.1, 0.2, and 0.3 s, using Application 2. All scenarios were configured with 500 tasks, 100 IoT devices and 1 MEC server. Figure 6 shows that small task generation rates flood the network with tasks, rapidly consuming all local resources. The x-axis shows the execution time in seconds, and the y-axis indicates the number of tasks. Remembering, 50% are critical tasks, and 50% are regular tasks.

The critical tasks are immediately launched to IoT devices with a 0.05 s task generation rate. However, as IoT processing is slowest than CC and MEC server, the tasks have a slow time completion. In contrast, regular tasks are launched to MEC and CC. As a result, the local network reaches an overhead quickly. With a 0.1 s task generation rate, critical tasks are completed in a more balanced fashion in IoT devices. The MEC servers can execute more tasks in comparison with CC.

More critical tasks can be executed in MEC servers with 0.2 s of task generation rate. Thus, CC resources are little-used in comparison to before generation task rates. Task generation rates should be designed with a time interval that favors local resource usage. It may help reduce total costs depending on the application and the costs of each allocation policy, as with the 0.3 s generation task rate shown in Figure 6.

By configuring deadlines with very restrictive time limits, the experiments showed that critical tasks were canceled because the scheduler could not find an allocation policy to achieve task completion. Deadlines must be appropriately configured to have sufficient time allocation to process and complete the task correctly and avoid this behavior.

#### 6.3.4. Using the DVFS Technique

With DVFS enabled, total energy consumption decreased by 13.74%, while total time increased by 28.32% compared to DVFS off. This demonstrates the effectiveness of the proposed model and the scheduling algorithm in minimizing the total energy consumption. Although the whole the time may have been longer in the approach with DVFS, it is not a problem because tasks were completed within the time limit imposed by the deadline.

Challenges about the time complexity of the DVFS technique have already been discussed by Chen et al. [16]. Our model addresses this problem by limiting the possible voltage–frequency pairs used to calculate dynamic power for task execution, allowing results to be obtained in feasible execution time.

## 7. Conclusions

Energy and time reduction are mostly needed for environments where large volumes of data and mobile devices are connected to the Internet with restricted QoS requirements and battery limitations. The TEMS algorithm chooses the most suitable allocation options in the system, reducing energy waste and elapsed time. Adequate coefficients allowed a decrease of energy consumption up to 51.6% and an execution time reduction up to 86.6%, ending critical tasks inside the deadline. Thus, the system becomes more sustainable, and the user experience is more satisfying.

The experiments exhibit that MEC server energy consumption is more efficient for applications when occurs task offloading from local IoT to MEC high workload applications and consequently saves battery energy. This is explained because IoT is closer to MEC than cloud. However, when there are high data transfer rates and a high number of tasks, the local processing policy in IoT devices can reduce energy consumption up to 23%, with a decrease in the task execution time of 29%.

The use of MEC servers helps increase the battery life of the IoT devices and enables agile task execution. Moreover, using the DVFS technique caused exciting results, supporting the energy consumption decrease. This work allowed contributions such as the TEMS algorithm and combining data transmission to the cost model. The model also considers idle costs, data transmission rate interference, using the DVFS technique, and the interaction with the CC layer to provide computational resources whenever the local network becomes overloaded.

As future works, we can indicate the progression of the system cost model to more fine grain, with the insertion of new variables and new environments to explore applications in different scenarios such as industry, healthcare, aviation, and mining. Another consideration is to evaluate new IoT applications in general cases to improve real-time dynamic mechanisms. We will evaluate finer coefficient controls to achieve the minimum energy consumption and the measured consumption to compare our solution with real world systems.

## Figures and Tables

**Figure 2 sensors-21-02914-f002:**
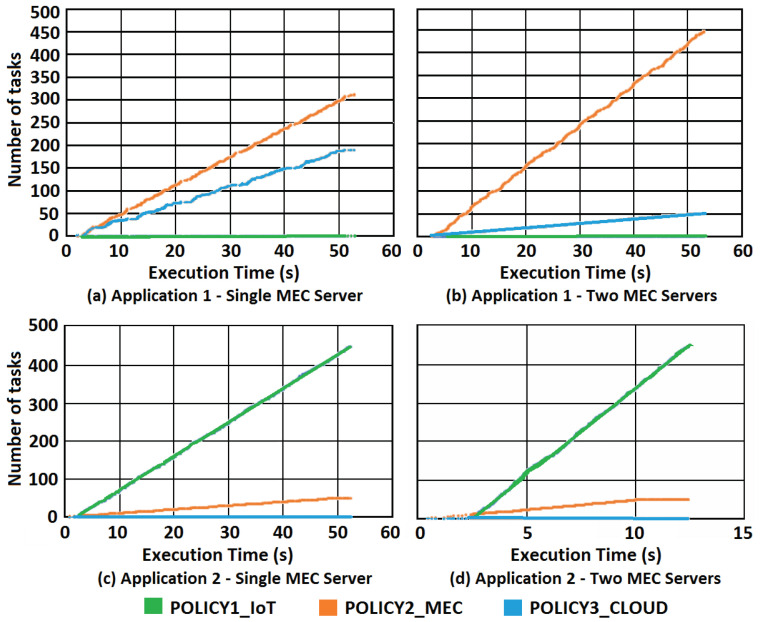
Task allocation for Application 1 and Application 2.

**Figure 3 sensors-21-02914-f003:**
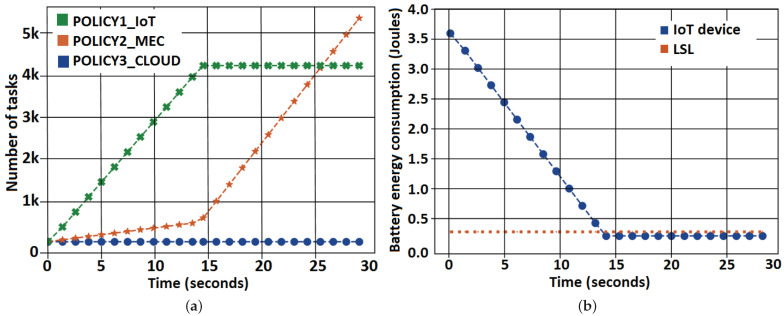
Task allocation behavior vs. IoT device battery energy consumption when the LSL is reached. (**a**)Task allocation in the system. (**b**) Battery energy consumption of IoT devices.

**Figure 4 sensors-21-02914-f004:**
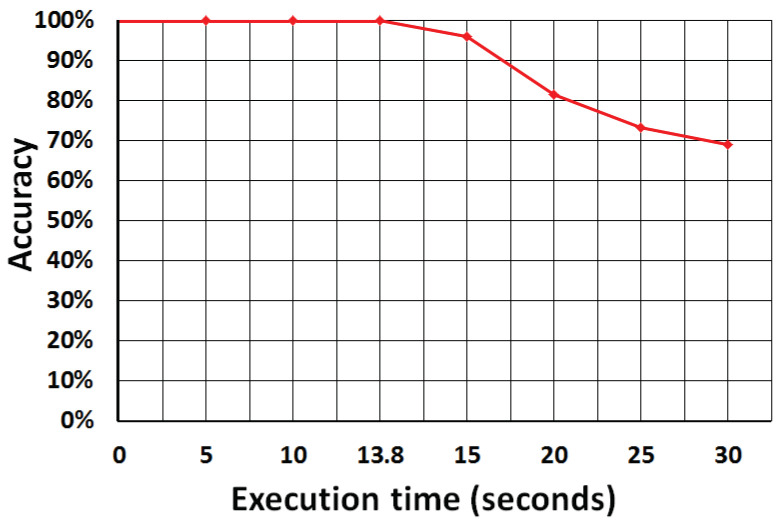
Analytic analysis of energy model accuracy.

**Figure 5 sensors-21-02914-f005:**
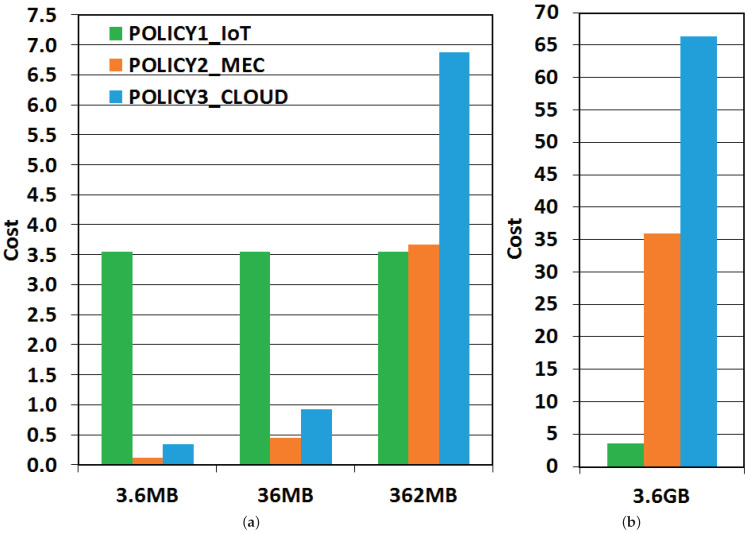
Cost policies for input data size variation in Application 1. (**a**) Cases A, B and C. (**b**) Case D.

**Figure 6 sensors-21-02914-f006:**
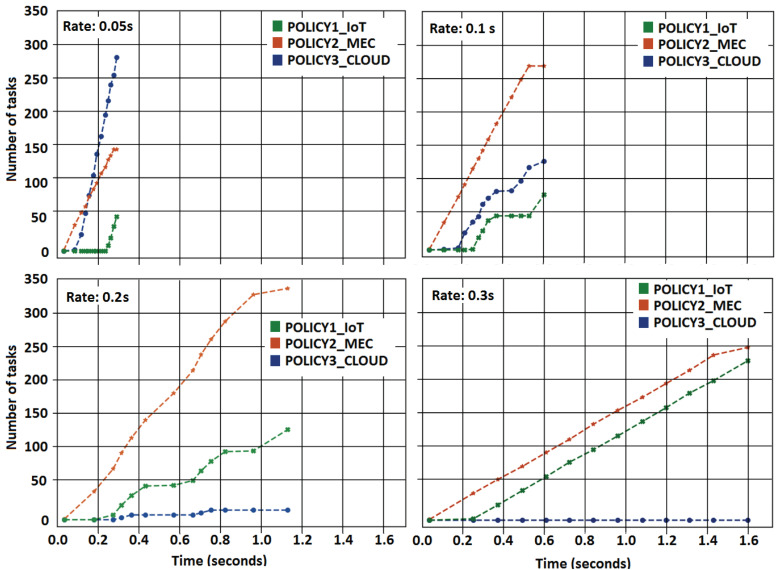
Task allocation for different task generation rates.

**Table 2 sensors-21-02914-t002:** Notation adopted for the model description.

#	Description	#	Description
*A*	The task set that will be executed.	*k*	An individual core.
*C*	Commutative capacitance.	*P*	Power consumed.
$C_{c}$	CPU cycles.	$P_{i,mec}$	The power consumed in the MEC server.
$Cc_{T}$	Total clock cycles.	PL	A CPU core set.
$Cost_{i,mec}$	The total cost in MEC server.	$pl_{j,n}$	A core *n* of a mobile device *j*.
*D*	A set of mobile IoT devices.	PS	A processor in the MEC server.
*d*	The input data.	*r*	Data transfer rate.
$d_{i}^{\prime }$	The return to the origin of results.	*S*	The number of MEC servers.
$E_{i,mec}$	The dynamic energy consumed in a MEC server by a task *i*.	sc	Source code offloading.
*E*	Energy consumption in idle time.	$S_{l}$	Local MEC server.
$f_{mec}$	Frequency of MEC server processor.	*t*	The deadline associated with the task.
*f*	Processor frequency.	*T*	Total execution time.
*H*	A set of the available wireless channels.	$T_{i,mec}$	Total execution time in the MEC server.
$h_{i}$	The wireless channel associated to task *i*.	$V_{local}$	Voltage in the IoT device processor.
*I*	Mutual interference rate.	$V_{mec}$	Voltage in the MEC server processor.
*i*	An individual task.	$W_{C}$	The number of wireless channels.
*j*	An individual mobile device.		

**Table 3 sensors-21-02914-t003:** Device variables for power calculation.

Hardware	Voltage-Frequency Pairs	Capacitance
IoT device	(5 V–16 MHz), (4 V–8 MHz), (2,7 V–4 MHz), (2.3 V–2 MHz), (1.8 V –1 MHz)	2.2 nF
MEC Server	(1.2 V–1500 MHz), (1 V–1000 MHz), (0.825 V–750 MHz), (0.8 V–600 MHz)	1.8 nF

**Table 4 sensors-21-02914-t004:** Characteristics of chosen applications.

Characteristics	Application 1	Application 2
Task generation rate (s)	10	0.1
Input Data (MB)	36.3	4
Result data size (bytes)	1250	625
Computational workload (Millions of CPU cycles)	2000	20
Critical tasks (% from total tasks)	10	50
Deadline for critical tasks (milliseconds)	500	100

**Table 5 sensors-21-02914-t005:** The MEC server benefit related to the energy consumption.

Cases Variables	A One MEC	B Two MEC	C W/o MEC
$E_{CPU}\left(J\right)$	2752.26	1725.18	5074.35
$E_{Trans}\left(s\right)$	835.60	1725.17	1166.21
$E_{mec}\left(J\right)$	3587.86	3450.35	6240.56
$T_{CORE}\left(s\right)$	956.21	1225.64	347.07
$T_{Trans}\left(s\right)$	199.75	159.69	290.31
$T_{TOTAL}\left(s\right)$	1155.96	1385.33	637.38
$T_{Trans}=T_{mec-up}+T_{mec-down}$			

**Table 6 sensors-21-02914-t006:** Cost coefficients for energy and time variation for Application 2.

Case	$u_{E}$	$u_{T}$	Cost $10^{-3}$	$E_{Total}$ mJ	$T_{Total}$ ms	Freq. MHz	Voltage V	Policy
C1	1/5	4/5	18.59	145.50	33.36	1,500	1.20	MEC
C2	2/5	3/5	25.97	142.76	34.69	750	0.83	MEC
C3	3/5	2/5	33.18	142.76	34.69	750	0.83	MEC
C4	4/5	1/5	35.44	70.40	250.00	8	4.00	IoT

## Data Availability

Not applicable.

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
