# Peer review of "An Algorithm to Minimize Energy Consumption and Elapsed Time for IoT Workloads in a Hybrid Architecture"

_sensors, 2021, doi:10.3390/s21092914_

Round 1
Reviewer 1 Report
The work proposes a dynamic cost model to minimize energy consumption and task processing time for IoT scenarios in Mobile Edge Computing environments. The work can publish subject to the following modifications:
1) The explanation is not very easy to read in language and the design of the paper.
2) Please use vector size graph representation for the plots/graphs/topologies: pdf/eps to have a higher quality plot/model presentation.
3)Can we have numerical results for 3 MEC servers (sequential)? Also, can we have an analytical closed-form solution for the k MEC services in this model?
4) The background needs to be enhanced using related state-of-the-art methods like P-SEP (Supercomputing) and N-SEP. Such methods apply the same to the IoT system dealing with MEC/Fog system that can give a better idea for the related functions.
Author Response
Reviewer: 1
Comments:
English language and style
(x) English language and style are fine/minor spell check required
The work proposes a dynamic cost model to minimize energy consumption and task processing time for IoT scenarios in Mobile Edge Computing environments. The work can publish subject to the following modifications:
Our considerations (Authors’ Answer)
Dear Reviewer,
We want to apologize for our text appears not being straightforward to read in language and the paper's design. We have reworked the text to remove possible problems. We hope that this new text changes the image of our work. We reviewed all sections carefully and fixed the found mistakes.
Q1) The explanation is not very easy to read in language and the design of the paper.
We reviewed all sections carefully and fixed the found mistakes. We introduced subsection 4.3 General Energy Consumption and a table called Symbols. Table 2 - Notation adopted for the model description was put closer to the model for easier understanding. Also, we rewrote the network model and TEMS algorithm text, review all formulas and our evaluations.
Q2) Please use vector size graph representation for the plots/graphs/topologies: pdf/eps to have a higher quality plot/model presentation.
All Figures were resized and updates. The tables in the text were improved and adjusted.
Q3) Can we have numerical results for 3 MEC servers (sequential)? Also, can we have an analytical closed-form solution for the k MEC services in this model?
It is not trivial to establish an analytical model for this issue because the MEC servers are suitable mainly when there are applications with high workloads. The system explores the offloading from IoT to MEC. However, when the application has hard deadline requirements for task executions, the best performances, and energy consumption saving in IoT devices are better achieved when it avoid offloading data. Therefore, there is no direct relationship between the number of MEC servers and different application types as in mission-critical applications, for instance. Thus, the analytic model could not represent an entire solution. Please, see Figure 2. Task allocation for Application 1 and Application 2. Because of this, we do not establish an analytic representation of the model.
Q4) The background needs to be enhanced using related state-of-the-art methods like P-SEP (Supercomputing) and N-SEP. Such methods apply the same to the IoT system dealing with MEC/Fog system that can give a better idea for the related functions.
The state-of-the-art was improved with the following text, and we updated Table 1 proposal comparison.
Naranjo et al. [29] propose an energy maximizing solution to prolong the aliveness of the wireless sensor networks with a Prolong Stable Election Protocol (P-SEP) in Fog infrastructure, decreasing energy consumption. An algorithm considers the distance between a cluster head (grouping of IoT sensors) and fog nodes to achieve load-balancing.
[29] Naranjo, P.G.V.; Shojafar, M.; Mostafaei, H.; Pooranian, Z.; Baccarelli, E. P-SEP: a prolong stable election routing algorithm for energy-limited heterogeneous fog-supported wireless sensor networks. The Journal of Supercomputing 2017,73, 733–755, doi:10.1007/s11227-016-1785-9.
Reviewer 2 Report
The DVFS methodology proposed by the authors clearly demonstrated the reduction in energy consumption. It could be observed that the task has been expedited with the utilization of MEC servers further enabling the life span of battery and IoT devices. The work has incorporated TEMS algorithm as well. Significant contributions has been shown by the authors in terms of energy consumption and task completion.
The paper is written well and I am recommending the paper towards acceptance since the objective has been clearly met.
Author Response
Reviewer: 2
Comments:
English language and style
(x) English language and style are fine/minor spell check required
The DVFS methodology proposed by the authors clearly demonstrated the reduction in energy consumption. It could be observed that the task has been expedited with the utilization of MEC servers further enabling the life span of battery and IoT devices. The work has incorporated TEMS algorithm as well. Significant contributions has been shown by the authors in terms of energy consumption and task completion.
The paper is written well and I am recommending the paper towards acceptance since the objective has been clearly met.
Our considerations (Authors’ Answer)
Dear Reviewer,
We would like to thank the comments and inform you that we reviewed all sections carefully to fix English.
Reviewer 3 Report
The paper provides a general model and algorithm to reduce the energy dissipated by
a wirelessIoT device in its communication and processing tasks.
The addressed topic is relevant, especially in the Mobile Edge Computing context, where
the dissipation of energy is a major problem. The proposed approach is intended to cover
both the energy consumed in the node processing activities and the one due to data transmissions.
Scheduling issues and occupied time are also considered in the described algorithm.
One strong point in the presented work is the complete software simulator that the authors
developed and made available on GitHub. Although the quality of the English writing needs to be
improved, the paper is well structured with a nice flow that leads the reader from the introduction
and the review of the state of the art up to the details of the proposed method and the related results.
One element that is missing at least in the review of the state of the art, and maybe also in the
subsequent modeling development, is the contribution of channel coding and error correction. Since the
work intends to propose as a complete and accurate model, I believe that the channel encoding/decoding,
which is a dominant component of the physical layer in terms of complexity and energy consumption, cannot
be neglected completely. Some papers appeared in both MDPI and IEEE journals on the use of
turbo and LDPC codes in wireless sensor networks nodes to optimize power consumption and trade-off
processing and transmission energy.
A second major weak point is the loss of comparisons with the previous literature. The authors provide
some numerical examples, but comparisons are necessary to assess the value of the given contribution. How
accurate is the proposed energy consumption model? To answer this question, the authors should show
both the comsumption estimated by means of their model and the measured consumption deriving from a real
system. This kind of comparison is likely to be rather difficult, but it could be simplified by
reporting the data only for a few components, if the overall system is not available, or data can be
found in the literature.
In DVFS, the switching between different combinations of supply voltage and clock frequency has a
relevant latency. It is not clear from the paper if the author consider thus issue and how.
Final conclusions would be more meaningful if they were associated with a sort of guidelines on the
design choices that can lead to a minimum energy consumption in different types of scenarios.
The English definitely needs improvement to remove the frequent mistakes and to improve readability.
Author Response
Reviewer: 3
Comments:
English language and style
(x) Extensive editing of English language and style required
Comments and Suggestions for Authors
The paper provides a general model and algorithm to reduce the energy dissipated by a wireless IoT device in its communication and processing tasks.
The addressed topic is relevant, especially in the Mobile Edge Computing context, where the dissipation of energy is a major problem. The proposed approach is intended to cover both the energy consumed in the node processing activities and the one due to data transmissions. Scheduling issues and occupied time are also considered in the described algorithm. One strong point in the presented work is the complete software simulator that the authors developed and made available on GitHub.
Although the quality of the English writing needs to be improved, the paper is well structured with a nice flow that leads the reader from the introduction and the review of the state of the art up to the details of the proposed method and the related results.
Our considerations (Authors’ Answer)
Dear Reviewer,
We want to thank the comments and inform you that we review all sections carefully to fix English. Also, we update the text according to your recommendation as following.
Q1) One element that is missing at least in the review of the state of the art, and maybe also in the subsequent modeling development, is the contribution of channel coding and error correction. Since the work intends to propose as a complete and accurate model, I believe that the channel encoding/decoding, which is a dominant component of the physical layer in terms of complexity and energy consumption, cannot be neglected completely. Some papers appeared in both MDPI and IEEE journals on the use of turbo and LDPC codes in wireless sensor networks nodes to optimize power consumption and trade-off processing and transmission energy.
The state-of-the-art was improved with the following text:
Gautham et al. [30] analyzed an architecture to evaluate the code/decode of the communication channel. In this particular, TEMS determines the better communication channel between IoT devices and MEC servers. Therefore, the contribution of channel coding/decoding and error correction are included in the choice of the channel with the lower cost.
In the network model, we add the following statement to the text.
The TEMS algorithm will choose a channel with a lower energy consumption cost. If the task runs in the local device, it does not have an associated channel. Coding and decoding costs are built-in to the channel like in the approach [30].
[30] Gautham, T.S.V.; Thangaraj, A.; Jalihal, D. Common architecture for decoding turbo and LDPC codes. National Conference On Communications (NCC), 2010, pp. 1–5. doi:10.1109/NCC.2010.5430239.
Q2) A second major weak point is the loss of comparisons with the previous literature. The authors provide some numerical examples, but comparisons are necessary to assess the value of the given contribution. How accurate is the proposed energy consumption model? To answer this question, the authors should show both the consumption estimated by means of their model and the measured consumption deriving from a real system. This kind of comparison is likely to be rather difficult, but it could be simplified by reporting the data only for a few components, if the overall system is not available, or data can be found in the literature.
We agree that comparing real-world systems in a not controlled environment is hard to do. Mainly because of the simulated applications in this article. The applications to avoid vehicle-to-vehicle collision and to recognize the roads' vehicle plates were not applied in similar scenarios in the literature as in our proposal. Thus we do not have similar data to compare without reproducing the experiments in the real world. However, it isn't easy at this moment. Another consideration is the health pandemic situation today without lab access in the University. All these problems also justify the option use of simulation. Thus, trying to overcome and minimize these problems, a vast comparison between Related Work and our solution was shown in Table 1.
Moreover, we add this issue as future work in the Section: Conclusion, with the following text:
As future works, we can indicate the progression of the system cost model to more fine grain, with the insertion of new variables and new environments to explore applications in different scenarios such as industry, healthcare, aviation, and mining. Another consideration is to evaluate new IoT applications in general cases to improve real-time dynamic mechanisms. Also, we will evaluate finer coefficient controls to achieve the minimum energy consumption and the measured consumption to compare our solution with real-world systems.
Q3) In DVFS, the switching between different combinations of supply voltage and clock frequency has a relevant latency. It is not clear from the paper if the author consider thus issue and how.
We clarify in the text, in Subsection: 4.4 Local Computing in the IoT Device with the following statement.
The DVFS associated overhead rate is between 0.02% to 2% in the best and worst-case scenarios, according to [16]. The cost overhead for this approach represents 2% in our model, built-in in these trade-off coefficients.
[16] Chen, Y.L.; Chang, M.F.; Yu, C.W.; Chen, X.Z.; Liang, W.Y. Learning-Directed Dynamic Voltage and Frequency Scaling Scheme with Adjustable Performance for Single-Core and Multi-Core Embedded and Mobile Systems. Sensors 2018, 18, 3068. doi:10.3390/s18093068.
Q4) Final conclusions would be more meaningful if they were associated with a sort of guidelines on the design choices that can lead to minimum energy consumption in different types of scenarios.
The suggestion was added to the conclusion text as following:
The experiments exhibit that MEC server energy consumption is more efficient for applications when occurs task offloading from local IoT to MEC high workload applications and consequently saves battery energy. This is explained because IoT is closer to MEC than Cloud. However, when there are high data transfer rates and a high number of tasks, the local processing policy in IoT devices can reduce energy consumption up to 23%, with a decrease in the task execution time of 29%.
Q5) The English definitely needs improvement to remove the frequent mistakes and to improve readability.
We reviewed all sections carefully, updated the text organization, and fixed the found mistakes.
Reviewer 4 Report
The paper presents TEMS algorithm to chooses the most suitable allocation options in order to reducing energy waste and elapsed time. Although this topic is not a hot new topic, the authors did not consider any new interesting parameters. Besides, the paper needs to modify the following items to improve the quality:
- In 4.2. Network Model, the author defined a Task including source code (sc) and input data (d) which are needed to be sent to the application processing. Why source code needed to be sent ? Could the authors cite or describe any real application? And please explain why "i" is defined here.
- In 4.3 Local Computing in the IoT Device, PL is defined as number of CPU core, it seems not relevant, what mean "j" here, and there is no detail explanation for P & T, it is really difficult to follow your equations.
- In evaluation section, the IoT applications are not well-described.
- For the coefficient parameter, it seems suitable for the test scenario the authors mentioned, but may not the best solution for another applications. The author should consider various IoT applications in general case, TEMS should be improved to support real time mechanism to dynamic control the coefficient to obtain the minimum energy consumption.
Author Response
Reviewer: 4
Comments:
The paper presents TEMS algorithm to chooses the most suitable allocation options in order to reducing energy waste and elapsed time. Although this topic is not a hot new topic, the authors did not consider any new interesting parameters. Besides, the paper needs to modify the following items to improve the quality:
Our considerations (Authors’ Answer)
We reviewed all sections carefully, updated the text organization, and fixed the found mistakes.
Q1) In 4.2. Network Model, the author defined a Task including source code (sc) and input data (d) which are needed to be sent to the application processing. Why source code needed to be sent ? Could the authors cite or describe any real application? And please explain why "i" is defined here.
We rewrote the text of Subsection 4.2: Network Model. We also add new Subsection 4.3: General Energy Consumption to turn more clear our model and easier the reading. An example was included following your suggestion in the text as:
Offloading is an advantage in industrial applications to reduce the congestion of data transmission and save energy consumption [13].
Q2) In 4.3 Local Computing in the IoT Device, PL is defined as number of CPU core, it seems not relevant, what mean "j" here, and there is no detail explanation for P & T, it is really difficult to follow your equations.
We included Table 2, with a notation adopted for the model description. Now it is closer to the model description. We adjusted the description of the main number to make the text clearer. Also, we review the meaning of the referred terms. We also add new Subsection 4.3: General Energy Consumption to turn more clear our model and adjust the text to Subsection: Local Computing in the IoT Device for easier reading.
Q3) In evaluation section, the IoT applications are not well-described.
The IoT applications were more detailed with the following text:
The simulated applications are two vehicular applications that were described originally in Jansson's Ph.D. Thesis [36]. Application 1 represents the image recognition of vehicle registration plates on roads. It has a high workload and a high task creation rate. Application 2 is a Vehicle-To-Vehicle communication to avoid car collisions. It is a critical-mission application with a low workload, low data, and a hard deadline. Because of this, the task generates rates are faster, in comparison with Application 1. Application 2 creates more tasks than Application 1 for the same interval time, but each task has lower processing requirements.
Q4) For the coefficient parameter, it seems suitable for the test scenario the authors mentioned, but may not the best solution for another applications. The author should consider various IoT applications in general case, TEMS should be improved to support real time mechanism to dynamic control the coefficient to obtain the minimum energy consumption.
This suggestion was considered as future work in the conclusion with the following text:
Another consideration is to evaluate more IoT applications in general cases to improve the support of real-time dynamic mechanisms for the coefficient controls to achieve the minimum energy consumption.
Round 2
Reviewer 1 Report
The paper is well updated, easy to read, has solid results, and supportive explanations provided. The reviewer has no further comments and it can be published in the current form.
Author Response
The paper is well updated, easy to read, has solid results, and supportive explanations provided. The reviewer has no further comments and it can be published in the current form.
Reviewer 3 Report
The authors have improved the quality of the paper. The review of the state of the art has been extended by including [30], which dates back to 2010. More recent works appeared in Sensors on this subject: maybe the authors can extend a bit more their review. As for the comparisons, they added some elements in Table 1. It would be nice to also comment on the accuracy of the energy model. The English quality was improved.
Author Response
Reviewer 3
Comments and Suggestions for Authors
The authors have improved the quality of the paper.
The review of the state of the art has been extended by including [30], which dates back to 2010. More recent works appeared in Sensors on this subject: maybe the authors can extend a bit more their review.
As for the comparisons, they added some elements in Table 1.
It would be nice to also comment on the accuracy of the energy model. The English quality was improved.
Our considerations (Authors’ Answer)
Dear Reviewer,
We want to thank you for your comments that helping us to improve this article.
Q1) The review of the state of the art has been extended by including [30], which dates back to 2010. More recent works appeared in Sensors on this subject: maybe the authors can extend a bit more their review. As for the comparisons, they added some elements in Table 1.
We added two more references [30] and [31] in Section: Related Work with the following text, and we added more elements to Table 1 for the comparisons adjusts, as your suggestion.
5G networks introduce designs and adaptable support to new applications. However, it requires latency management, high energy efficiency, and long-range communication support for IoT-based applications [31]. Mucchi et al. [30] propose to add a physical layer to the burst data transmission management with the insertion of a zero-energy symbol for the wireless transmission when there is the send of discontinuous data (silence periods). As a result, the system saves energy consumption from IoT devices. In this approach, the silence produces a fine-grain granularity to energy management. The algorithms optimize power consumption between processing and energy spent on transmission.
[30] Mucchi, L.; Ronga, L.S.; Jayousi, S. Energy Efficient Constellation for Wireless Connectivity of IoT Devices. Sensors 2020, 20, 3991:1–3991:15. doi:10.3390/s20143991.
[31] Ahad, A.; Tahir, M.; Aman Sheikh, M.; Ahmed, K.I.; Mughees, A.; Numani, A. Technologies Trend towards 5G Network for Smart Health-Care Using IoT, A Review. Sensors 2020, 20, 4047:1–4047:22. doi:10.3390/s20144047.
The adjusted reference in the previous revision was changed to [29].
Q2) It would be nice to also comment on the accuracy of the energy model.
We added subsection 6.3 Accuracy Evaluation of Energy Model with an analytic analysis of model accuracy, trying to overcome this issue.
Reviewer 4 Report
Thank you for your response and consider to enhance content according to my recommendation. The paper looks better now for publication.
Author Response
Dear Reviewer,
We want to thank you for your comments that helping us to improve this article.